# Prognostic Significance of Programmed Cell Death Ligand 1 Expression in High-Grade Serous Ovarian Carcinoma: A Systematic Review and Meta-Analysis

**DOI:** 10.3390/diagnostics13203258

**Published:** 2023-10-19

**Authors:** Jeongwan Kang, Kang Min Han, Hera Jung, Hyunchul Kim

**Affiliations:** Department of Pathology, CHA Ilsan Medical Center, 1205 Jungang-ro, Ilsandong-gu, Goyang-si 10414, Republic of Korea; kang210@chamc.co.kr (J.K.); kiekie53@chamc.co.kr (K.M.H.);

**Keywords:** PD-L1, high-grade serous carcinoma, ovary cancer, prognosis, biomarker

## Abstract

(1) Background: High-grade serous ovarian carcinoma (HGSOC) is an aggressive subtype of ovarian cancer. Recent advances have introduced prognostic markers and targeted therapies. Programmed cell death ligand 1 (PD-L1) has emerged as a potential biomarker for HGSOC, with implications for prognosis and targeted therapy eligibility; (2) Methods: A literature search was conducted on major databases, and extracted data were categorized and pooled. Subgroup analysis was performed for studies with high heterogeneity. (3) Results: Data from 18 eligible studies were categorized and pooled based on PD-L1 scoring methods, survival analysis types, and endpoints. The result showed an association between high PD-L1 expression and a favorable prognosis in progression-free survival (HR = 0.53, 95% CI = 0.35–0.78, *p* = 0.0015). Subgroup analyses showed similar associations in subgroups of neoadjuvant chemotherapy patients (HR = 0.6, 95% CI = 0.4–0.88, *p* = 0.009) and European studies (HR = 0.59, 95% CI = 0.42–0.82, *p* = 0.0017). In addition, subgroup analyses using data from studies using FDA-approved PD-L1 antibodies suggested a significant association between favorable prognosis and high PD-L1 expression in a subgroup including high and low stage data in overall survival data (HR = 0.46, 95% CI = 0.3–0.73, *p* = 0.0009). (4) Conclusions: This meta-analysis revealed a potential association between high PD-L1 expression and favorable prognosis. However, caution is warranted due to several limitations. Validation via large-scale studies, with mRNA analysis, whole tissue sections, and assessments using FDA-approved antibodies is needed.

## 1. Introduction

High-grade serous ovarian carcinoma (HGSOC) stands as the most common histopathological subtype of ovarian cancer, characterized by its aggressive behavior [1]. A significant portion of patients are diagnosed at an advanced stage, with a reported median 5-year survival rate ranging from 15 to 55 percent [2]. Recent advancements have brought forth prognostic markers, including the homologous recombination deficiency (HRD) test and BRCA1/2 mutation analysis [3,4]. Importantly, these tests not only provide prognostic insights but also assist in identifying eligible candidates for targeted therapy with poly ADP ribose polymerase (PARP) inhibitors, a recent addition to the arsenal of anticancer drugs [5,6].

Programmed cell death ligand 1 (PD-L1), a potential biomarker for HGSOC, serves as a valuable indicator for prognostication and eligibility assessment for targeted therapies in select cancer types [7]. PD-L1 functions by binding to its receptor, PD-1, triggering apoptosis in activated T cells, thereby enabling tumor cells to evade host immune responses via this mechanism [8]. Research into the prognostic significance of PD-L1 expression is expanding, accompanied by a broadening application of PD-L1 inhibitors [9]. Nevertheless, the precise prognostic implications of PD-L1 expression in ovarian cancer remain elusive. In the context of HGSOC, several studies have explored the association between PD-L1 expression and prognosis [10,11,12,13,14,15,16,17,18,19,20,21,22,23,24,25], and meta-analyses on PD-L1 expression in ovarian cancer have been conducted [26,27,28,29]. However, both individual studies and meta-analyses have yielded inconsistent and contradictory results, primarily due to the amalgamation of data from various histological types and limited subject numbers.

To gain deeper insights into the prognostic relevance of PD-L1 expression in HGSOC, we conducted a comprehensive meta-analysis. This analysis involved meticulous data extraction, rigorous stratification, and stringent analytical criteria, all aimed at elucidating the survival outcomes of HGSOC patients in relation to PD-L1 expression.

## 2. Materials and Methods

### 2.1. Literature Search and Study Design

We conducted a comprehensive literature search on 29 June 2023 on PubMed, Embase, Web of Science, MEDLINE, and the Cochrane Library. Our search strategy included the keywords “PD-L1” and “high-grade serous carcinoma”. Subsequently, two authors (H.J. and H.K.) independently evaluated the search results. This study adhered to the Preferred Reporting Items for Systematic Reviews and Meta-analyses (PRISMA) guidelines [30].

### 2.2. Literature Selection

We selected studies that examined associations between prognosis and PD-L1 expression via immunohistochemistry (IHC). The following exclusion criteria were applied: (1) Obviously irrelevant articles. (2) Conference abstract without sufficient data. (3) Review articles, case studies, letters, or errata. (4) Non-English articles. (5) Studies using evaluation methods other than IHC (e.g., polymerase chain reaction (PCR) or ELISA). (6) Article with insufficient or irrelevant data. (7) Newcastle–Ottawa scale 5 or less. In cases where multiple articles employed overlapping patient populations, we opted to include only the most informative article. Notably, some studies raised the possibility of overlapping patient populations [13,19,20,23]. However, these studies were integrated into our analysis due to their utilization of distinct evaluation methods and data types, which mitigated concerns about redundancy and provided valuable insights.

Two authors (H.J. and H.K.) applied the Newcastle–Ottawa Quality Assessment Scale (NOS) for the assessment of the quality of the included studies (Appendix A). Any disagreements that arose during the assessment were resolved via discussion.

### 2.3. Extraction of Data

Two authors (H.J. and H.K.) independently extracted the following information: authors, year of publication, source of the study population, study region, PD-L1 antibody clone and manufacturer, slide type (whole section VS. tissue microarray), PD-L1 IHC scoring method, the cancer stage of the study population, the inclusion of neo-adjuvant chemotherapy (NACT) patients, and survival data (type of data, hazard ratio (HR), confidence interval, *p*-value, and the number of patients of control and experimental groups).

In instances where specific data from the Kaplan–Meier curve were not present, we computed the hazard ratio (HR) using data extracted by the Engauge Digitizer software, version 12.1, following the methodology outlined by Irvine et al. [31]. When we had access to the HR, the number of patients in each group, and the confidence interval along with the *p*-value, we calculated the standard error utilizing the approach established by Tierney et al. [32].

### 2.4. Statistical Analysis

Statistical analyses were conducted with R, version 4.3.0 [33], and the “meta” package, version 6.2.1 [34]. To provide a comprehensive assessment, we categorized survival data based on PD-L1 scoring methods (tumor cells (TC), combined positive score (CPS), and immune cells (IC)), survival analysis types (Kaplan–Meier estimates and multivariate or univariate Cox proportional hazard models), the specific survival endpoint (overall survival (OS)), and progression-free survival (PFS).

To gauge the prognostic significance of PD-L1 expression, we pooled the hazard ratios (HRs) from studies employing the same PD-L1 scoring method, survival analysis type, and endpoint, along with their corresponding standard errors. Heterogeneity was assessed using the I^2^ value, with an I^2^ value exceeding 50% indicating substantial heterogeneity. We employed a random-effects model for our analysis to account for this heterogeneity. Subgroup analysis was conducted when significant heterogeneity was observed, aiming to refine our pooled estimates.

To enhance the robustness of our findings, we conducted additional analyses specifically focusing on studies employing U.S. Food and Drug Administration (FDA) approved PD-L1 antibodies. In this analysis, the FDA-approved PD-L1 antibodies were an SP263 clone from Ventana (Roche) and a 22C3 clone from Dako.

Egger’s test, Begg’s test, and funnel plot for HRs were performed in analyses comprising more than seven studies. Additionally, sensitivity analysis was carried out to assess the stability of our results by systematically excluding one study at a time from the pooled analysis. *p*-values less than 0.05 were considered statistically significant.

## 3. Results

### 3.1. Characteristics of the Included Studies

The literature selection process is shown in Figure 1. Out of the 307 initially retrieved studies, 135 duplicate articles were removed. A number of 172 studies were screened, and 126 were excluded (80 conference abstracts with insufficient information, 34 obviously irrelevant articles, 6 review articles, 4 case reports, 1 non-English article, and 1 study with methods other than IHC). The full text was retrieved for the remaining 46 studies and 28 studies were excluded (27 studies with insufficient data and 1 study with methods other than IHC). A total of 18 studies were used for this meta-analysis.

Table 1 shows a summary of the studies included in this meta-analysis. The included studies were published from 2015 to 2023. Regions of the study populations were Europe (seven studies), Asia (six studies), America (two studies), and Africa (one study). Eight studies used tissue microarray (TMA), and six studies used whole tissue section slides. The antibody clones used for the included studies are as follows: 22C3 (five studies), SP263 (four studies), E1L3N (two studies), SP142 (one study), GB11339 (one study), and GR1 (one study). The antibodies were mostly rabbit monoclonal (nine studies), some were mouse monoclonal (five studies), and one study used rabbit monoclonal antibody. For the PD-L1 scoring method, 11 studies reported TC data, 6 studies reported CPS data, and 4 studies reported IC data. Many of the studies (15 studies) used Kaplan–Meier estimate survival analysis, and 4 studies used the Cox proportional hazard model. OS (15 studies), PFS (4 studies), disease-free survival (DFS) (2 studies), and recurrent-free survival (RFS) (1 study) were adopted for survival data endpoints in the included studies. DFS and RFS data could not be used for this meta-analysis because only one result was present in each group of the same survival analysis and the same endpoint.

Table 2 provides a summary of the grouped data for our prognostic meta-analysis, categorized by PD-L1 scoring method, survival analysis type, and survival endpoint.

### 3.2. Prognostic Significance of PD-L1 Expression

Table 3 and Figure 2 show summaries of meta-analysis of each survival data group. There were five groups of survival data available for analysis. OS with CPS scoring and Kaplan–Meier estimates was present in five studies. The combined HR was not statistically significant (HR = 0.9, 95% CI = 0.64–1.26, *p* = 0.53) without substantial heterogeneity (I^2^ = 54.1%, *p* = 0.07). OS with IC scoring and a multivariate Cox proportional hazard model was present in three studies. The combined HR was not significant (HR = 0.82, 95% CI = 0.54 = 1.26) without substantial heterogeneity (I^2^ = 33.7, *p* = 0.22). OS with TC scoring and a multivariate Cox proportional hazard model was present in four studies. The combined HR was not significant (HR = 0.93, 95% CI = 0.54–1.62) with substantial heterogeneity (I^2^ = 73.5%, *p* = 0.01). OS with TC scoring and Kaplan–Meier estimates was present in eight studies. The combined HR was not significant (HR = 0.9, 95% CI = 0.48–1.69) with substantial heterogeneity (I^2^ = 86.5%, *p* < 0.0001). PFS with TC scoring and Kaplan–Meier estimates was present in three studies. The combined HR was statistically significant (HR = 0.53, 95% CI = 0.35–0.78) without substantial heterogeneity (I^2^ = 0, *p* = 0.42).

### 3.3. Subgroup Analysis

Subgroup analysis was conducted on two data groups: OS data with CPS scoring and Kaplan–Meier estimates and OS data with TC scoring and Kaplan–Meier estimates (Table 4).

Subgroup analysis of OS data with CPS scoring and Kaplan–Meier estimates was carried out based on FDA approval of the antibody, stage, NACT, and antibody species. Regarding subgroup analysis by NACT, the NACT-included studies showed HR with statistical significance (HR = 0.6, 95% CI = 0.4–0.88, *p* = 0.009) without substantial heterogeneity (I^2^ = 0, *p* = 0.81). The rest of the subgroup analysis showed no significant results.

Subgroup analysis of OS data with TC scoring and Kaplan–Meier estimates was carried out based on FDA approval, stage, NACT, antibody species, region, and slide type. Regarding subgroup analysis by region, the European studies showed HR with statistical significance (HR = 0.59, 95% CI = 0.42–0.82, *p* = 0.0017) without substantial heterogeneity (I^2^ = 5.4, *p* = 0.3). The rest of the subgroup analysis showed no significant results.

### 3.4. Subgroup Analysis of Data from Studies with FDA-Approved Antibodies

Two data groups were available for subgroup analysis with FDA approved antibodies (Table 5).

OS data with CPS scoring and Kaplan–Meier estimates were available for subgroup analysis by stage. The result was not significant.

OS data with TC scoring and Kaplan–Meier estimates were available for subgroup analysis by stage, NACT, and antibody species. Regarding subgroup analysis by stage, studies with both high- and low-stage data showed significant results (HR = 0.46, 95% CI = 0.3–0.73, *p* = 0.0009) without substantial heterogeneity (I^2^ = 49.3, *p* = 0.13). The rest of the results were not significant.

### 3.5. Publication Bias

The tests for publication bias were conducted for HRs of OS with TC scoring and Kaplan–Meier estimates. Statistically significant publication bias was not seen in Egger’s test (*p* = 0.5) and Begg’s test (*p* = 0.62). The funnel plot is shown in Figure 3.

The sensitivity analysis of OS data with TC scoring and Kaplan–Meier estimates demonstrated robustness of the result and did not show the possibility of a high risk of bias originating from any specific studies (Figure 3).

## 4. Discussion

Our meta-analysis focused on investigating the prognostic significance of PD-L1 expression in HGSOC. Notably, this is the first meta-analysis solely dedicated to analyzing this association in HGSOC, and it involved extensive data extraction from relevant studies. A distinctive feature of our analysis was the meticulous classification of data based on their type and analysis method, enhancing the quality of the data. Additionally, we sought to enhance the robustness of our findings by conducting supplementary subgroup analyses, specifically considering studies that employed FDA-approved PD-L1 antibodies. Furthermore, we conducted subgroup analyses to delve deeper into the heterogeneity observed in the results, exploring factors that might influence the outcomes.

The PD-1/PD-L1 pathway is a well-established mechanism for immune evasion employed by malignant tumor cells [9]. PD-L1, a ligand for the PD-1 receptor [8], is commonly found in the microenvironment of malignant tumors in various organs, including the stomach, liver, lung, colon, breast, urinary bladder, and skin [35,36,37,38,39,40,41]. Within the tumor microenvironment, cells expressing PD-L1 include tumor cells, fibroblasts, and immune cells such as lymphocytes, macrophages, and dendritic cells [42]. When PD-L1 ligands bind to the PD-1 receptor on T lymphocytes, they transmit inhibitory signals, enabling tumor cells to evade host immunity effectively [9]. Beyond its role in immune evasion, PD-L1 also functions as a signaling molecule within the tumor microenvironment, promoting the secretion of inflammatory cytokines [42]. Due to these multifaceted roles, high levels of PD-L1 expression are believed to facilitate tumor proliferation and correlate with poorer prognosis in various malignancies [7]. Consequently, PD-L1 inhibitors have emerged as effective targeted anticancer drugs [9].

This meta-analysis unveiled a potential association between PD-L1 expression and a favorable prognosis in HGSOC. Specifically, when examining PFS with TC scoring and Kaplan–Meier estimates, a significant association was observed between PD-L1 expression and a favorable prognosis. This finding contradicts the more widely recognized association between high PD-L1 expression and worse prognosis observed in many cancers [7]. Interestingly, a positive correlation between high PD-L1 expression and a favorable prognosis has been reported in various cancers such as Merkel cell carcinoma, melanoma, colon cancer, stomach cancer, and small cell lung carcinoma [43,44,45,46,47].

However, studies exploring the relationship between PD-L1 expression and prognosis in HGSOC have yielded diverse results. Some studies have reported an association between PD-L1 expression and a worse prognosis, while others have found a link between PD-L1 expression and a favorable prognosis [10,11,12,13,14,15,16,17,18,19,20,21,22,23,24,25]. Previous meta-analyses on PD-L1 expression in ovarian cancers attempted to compare the prognostic associations between serous carcinoma and other types of carcinomas. Most of these studies could not establish significant results in serous carcinoma groups, but one study did find a significant association between PD-L1 expression and better prognosis in HGSOC [29]. Nevertheless, the results of meta-analyses focusing on ovarian cancers of other histological types demonstrated an association between worse prognosis and PD-L1 expression [26,27,29].

These contradictory findings in terms of prognostic significance have not gone unnoticed by researchers. They have contemplated innate confounding factors within the IHC study methods as potential reasons for these disparities [12,13,14,16,24,25]. However, even studies utilizing mRNA expression data have shown similar contradictions in the prognostic significance of PD-L1 expression in HGSOC [14,24,48]. Furthermore, Wieser et al. were unable to find a correlation between PD-L1 IHC data and mRNA data, suggesting the need for a more accurate and standardized method for IHC evaluation [49].

The association between a favorable prognosis and high PD-L1 expression is believed to be a consequence of adaptive immunity [24,25]. Specifically, researchers have pointed out the T cell subpopulation within the tumor microenvironment as a key determinant of prognostic significance [14]. They have observed an increased number of CD8-positive lymphocytes and PD-L1 staining at the invasive front of HGSOC [13,24]. The hypothesis put forward is that PD-L1 expression is a result of the tumor cell’s attempt to evade the host’s immune response, which in turn could lead to a favorable prognosis [14,23]. It is suggested that PD-L1 expression induced by adaptive immunity might be mediated via the interferon-gamma-related inflammatory pathway [17,50]. Using this mechanism driven by adaptive immunity, patients with PD-L1 expression may potentially benefit from PD-L1 inhibitors [23,24,25].

Our subgroup analysis aimed to address potential sources of heterogeneity, and within this rigorous examination, we observed a few significant associations between PD-L1 expression and favorable prognosis. NACT included a subgroup of OS with CPS scoring and Kaplan–Meier estimates, and the European subgroup of OS with TC scoring and Kaplan–Meier estimates showed a significant association between PD-L1 expression and favorable prognosis. Researchers investigating PD-L1 expression in HGSOC have frequently highlighted various factors as potential confounding elements, including differences in antibodies used, IHC staining protocols and equipment, scoring methods and interpreter subjectivity, cutoff values, slide types (whole tissue section vs. tissue microarrays), BRCA mutation status, sample sizes, tumor stages, and tumor heterogeneity [12,13,14,16,24,25]. In our subgroup analysis, we considered several factors that might contribute to heterogeneity. These factors included FDA approval of the antibodies, the inclusion of high-stage patients only, the use of neoadjuvant chemotherapy, the species of the antibody, geographical regions of the study, and slide types. However, it is important to reiterate that only a limited number of these subgroup analyses yielded significant results, and these results were based on a small number of studies. We specifically examined the antibody species used in IHC because different antibodies are known to produce varying IHC results [51]. We also looked at tumor stage as a potential factor since differences in characteristics between early and advanced-stage HGSOCs were reported [19,24]. NACT was considered due to its known influence on tumor-infiltrating lymphocytes and PD-L1 expression in ovarian cancers [52,53,54]. Geographical region was included as a factor because genetic and environmental differences can impact prognosis [23]. Additionally, we explored slide types used for IHC (whole tissue section slides vs. tissue microarrays) because PD-L1 is known to produce focal staining results [13,15,25], and tissue microarrays may not accurately represent PD-L1 expression due to tumor heterogeneity [13,15,25]. While there is evidence suggesting that the presence of BRCA1/2 gene mutations may lead to increased PD-L1 expression [13,21,23], we could not perform a subgroup analysis based on BRCA1/2 gene mutation status due to insufficient data.

We conducted an additional subgroup analysis utilizing data from studies that employed FDA-approved antibodies. We hypothesized that these studies might yield more consistent results. However, we observed that only the high and low stage group from the OS data with TC scoring and Kaplan–Meier estimates showed a significant association with favorable prognosis. This suggests that the application of FDA-approved antibodies might not uniformly impact the prognostic significance of PD-L1 expression across all subgroups.

While our analysis revealed several significant findings, all indicating a favorable prognosis associated with PD-L1 expression, it is crucial to approach these results with caution due to certain limitations in this meta-analysis. Notably, the data used in this meta-analysis was primarily derived from studies employing the IHC method. IHC results can be influenced by various factors such as the type and clone of antibodies used, the manufacturer, equipment, cutoff values, scoring methods, and the subjectivity of interpreters. Additionally, we estimated hazard ratios (HRs) from Kaplan–Meier curves. While we employed a relatively new HR estimation method, some level of discrepancy between actual HRs and estimated HRs is expected [31]. It is important to acknowledge that many of our analyses did not yield significant results and exhibited high heterogeneity. Furthermore, the significant results were often based on a relatively small number of studies, which could introduce some degree of bias. Another limitation is our restriction to English-language articles, potentially excluding valuable findings published in other languages.

## 5. Conclusions

This meta-analysis investigating the prognostic significance of PD-L1 expression in HGSOC has revealed a potential association between high PD-L1 expression and favorable prognosis. This finding suggests the potential utility of PD-L1 inhibitors in the treatment of HGSOC. However, it is crucial to exercise caution when interpreting these results due to the study’s limitations. The association between PD-L1 expression and a positive prognosis in HGSOC warrants further validation using large-scale studies employing both IHC and mRNA analysis methods. Additionally, comprehensive assessments using FDA-approved antibodies and scoring methods on whole tissue sections are essential to compare and confirm the prognostic value of this biomarker.

## Figures and Tables

**Figure 1 diagnostics-13-03258-f001:**
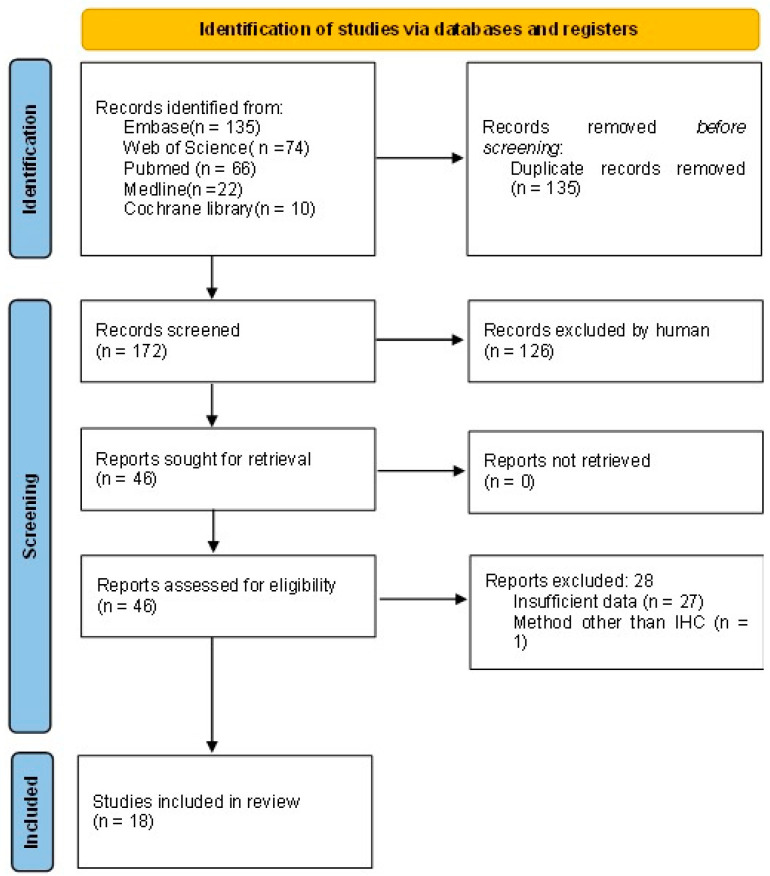
PRISMA flow diagram.

**Figure 2 diagnostics-13-03258-f002:**
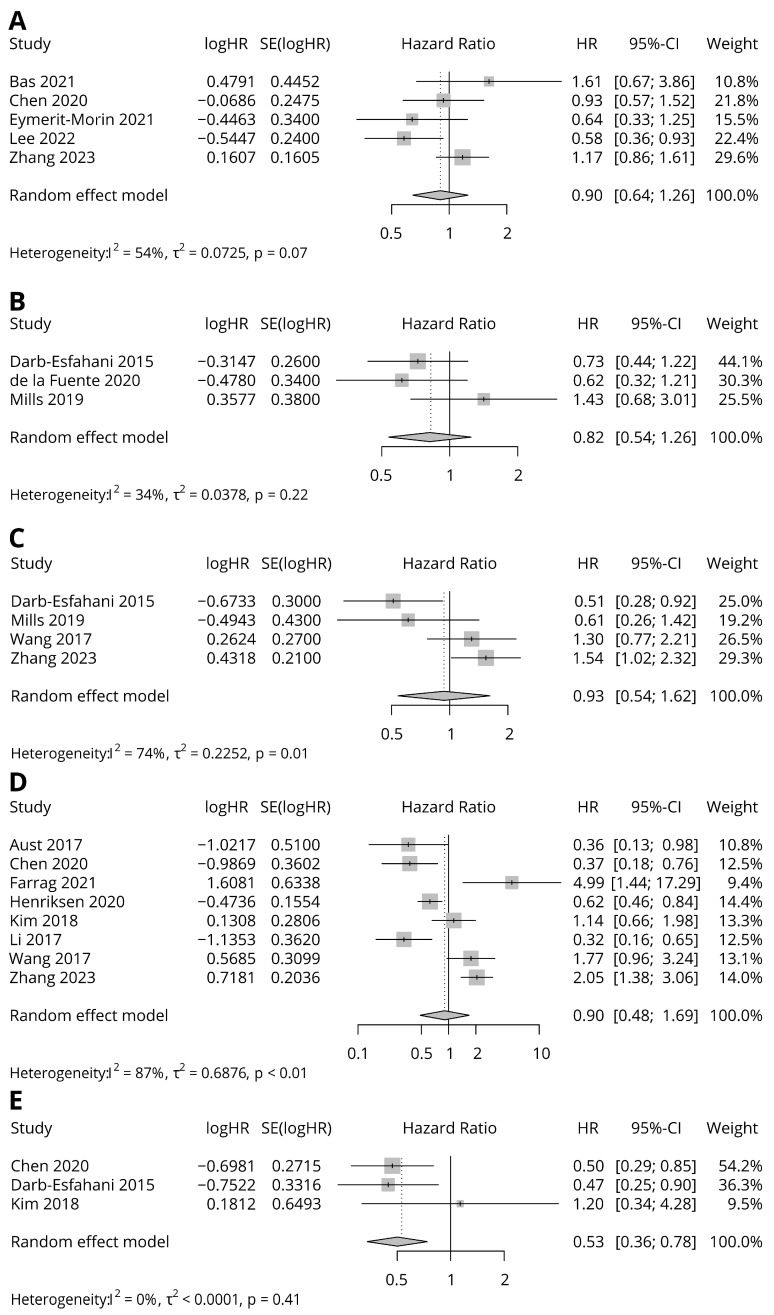
Forest plots for HRs in association with PD-L1 expression. (**A**) OS with CPS scoring and Kaplan–Meier estimate; (**B**) OS with IC scoring and multivariate Cox proportional hazard model; (**C**) OS with TC scoring and multivariate Cox proportional hazard model; (**D**) OS with TC scoring and Kaplan–Meier estimate; (**E**) PFS with TC scoring and Kaplan–Meier estimate [10,12,13,14,15,16,18,19,20,21,22,23,24,25].

**Figure 3 diagnostics-13-03258-f003:**
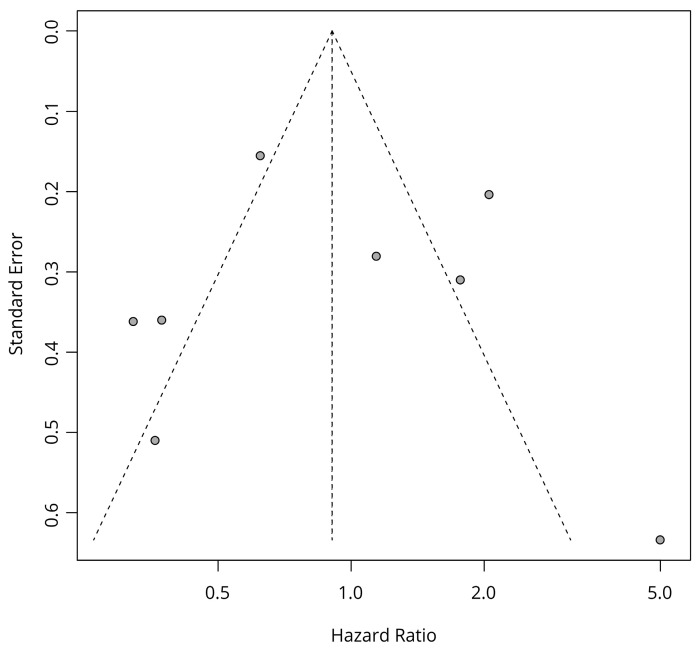
Funnel plot of OS data with TC scoring and Kaplan–Meier estimates.

**Table 1 diagnostics-13-03258-t001:** Characteristics of the included studies.

Study	Region	Slide Type	Ab ^1^ Clone	Ab ^1^ Manufacturer	Ab Species	Ab Clonality	Scoring Method	Survival Analysis Type	Endpoint
Aust 2017 [10]	Europe	Whole section	E1L3N	Cell Signaling	Rabbit	monoclonal	TC ^3^	KM ^6^	OS
Bansal 2021 [11]	Asia	TMA ^2^	SP263	Ventana (Roche)	Rabbit	monoclonal	CPS ^4^	KM ^6^	DFS
Bas 2021 [12]	Europe	TMA ^2^	22C3	Dako	Mouse	monoclonal	CPS ^4^	KM ^6^	OS
Chen 2020 [13]	America	Whole section	22C3	Dako	Mouse	monoclonal	TC ^3^	KM ^6^	OS, PFS
							CPS ^4^	KM ^6^	OS, PFS
Darb-Esfahani 2015 [14]	Europe	TMA ^2^	EPR1161	Abcam	Rabbit	monoclonal	TC ^3^	KM ^6^	PFS
								Cox ^7^ (multi ^9^)	OS, PFS
							IC ^5^	Cox ^7^ (multi ^9^)	OS, PFS
de la Fuente 2020 [22]	Europe	TMA ^2^	22C3	Dako	Mouse	monoclonal	IC ^5^	KM ^6^	OS
Eymerit-Morin 2021 [15]	Europe	TMA ^2^	QR1	Diagomics	Rabbit	monoclonal	CPS ^4^	Cox ^7^ (uni ^8^, multi ^9^)	OS, DFS
Farrag 2021 [16]	Africa	N/A	GB11339	Servicebio	Rabbit	polyclonal	TC ^3^	KM ^6^	OS, DFS
Fucikova 2019 [17]	Europe	Whole section	N/A	N/A	N/A	N/A	TC ^3^	KM ^6^	RFS
Henriksen 2020 [18]	Europe	N/A ^2^	SP263	Ventana (Roche)	Rabbit	monoclonal	TC ^3^	KM^6^	OS
Kim 2018 [19]	Asia	TMA ^2^	SP263	Ventana (Roche)	Rabbit	monoclonal	TC ^3^	KM ^6^	OS, PFS
Lee 2022 [20]	Asia	TMA ^2^	22C3	Dako	Mouse	monoclonal	CPS ^4^	KM ^6^	OS, PFS
Li 2017 [21]	Asia	TMA ^2^	SP263	Ventana (Roche)	Rabbit	monoclonal	TC ^3^	KM ^6^	OS
Mills 2019 [23]	America	Whole section	SP142	Spring Bioscience	Rabbit	monoclonal	TC ^3^, IC ^5^	KM ^6^	OS
Wang 2017 [24]	Asia	Whole section	E1L3N	Cell Signaling	Rabbit	monoclonal	TC ^3^	Cox ^7^ (multi ^9^)	OS
							TC ^3^, IC ^5^	KM ^6^, Cox (multi ^9^)	OS
Zhang 2023 [25]	Asia	Whole section	22C3	Dako	Mouse	monoclonal	TC ^3^	Cox ^7^ (uni ^8^)	OS
							TC ^3^	KM ^6^	RFS
							CPS ^4^	Cox ^7^ (uni ^8^, multi ^9^)	OS, RFS

^1^ Ab: antibody, ^2^ TMA: tissue microarray, ^3^ TC: tumor cell score, ^4^ CPS: combined positive score, ^5^ IC: immune cell score, ^6^ KM: Kaplan–Meier estimate, ^7^ Cox: Cox proportional hazard model, ^8^ uni: univariate analysis, ^9^ multi: multivariate analysis.

**Table 2 diagnostics-13-03258-t002:** Categorized data of the included studies.

Study	Scoring Method	Cut-Off	Survival Analysis Type	Survival Endpoint	PD-L1 Positive Rate	HR ^1^	SE ^2^ of ln(HR)	CI ^3^	High Stage Only	High Stage Proportion	NACT ^4^ Included	Slide Type	FDA Approval	Ab Species
Aust 2017 [10]	TC	1%	KM	OS	14/20	0.36	0.51	0.13~0.98	No	N/A	No	Whole section	Not approved	Rabbit
Chen 2020 [13]	TC	1%	KM	OS	21/100	0.37	0.36	0.18~0.76	No	91%	N/A	Whole section	Approved	Mouse
Farrag 2021 [16]	TC	1%	KM	OS	29/45	4.99	0.63	1.44~17.29	No	74%	Yes	N/A	Not approved	Rabbit
Henriksen 2020 [18]	TC	1%	KM	OS	164/283	0.62	0.16	0.46~0.84	No	83%	No	N/A	Approved	Rabbit
Kim 2018 [19]	TC	5%	KM	OS	36/108	1.14	0.28	0.65~1.98	Yes	N/A	Yes	TMA	Approved	Rabbit
Li 2017 [21]	TC	Custom method	KM	OS	13/112	0.32	0.36	0.16~0.65	No	85%	No	TMA	Approved	Rabbit
Wang 2017 [24]	TC	5%	KM	OS	26/81	1.77	0.31	0.96~3.24	No	76%	No	Whole section	Not approved	Rabbit
Zhang 2023 [25]	TC	1%	KM	OS	81/212	2.05	0.2	1.37~3.06	Yes	N/A	No	Whole section	Approved	Mouse
Darb-Esfahani 2015 [14]	TC	0 (any)	Cox (multi)	OS	178/202	0.51	0.3	0.28~0.91	No	N/A	No	TMA	Not approved	Rabbit
Mills 2019 [23]	TC	1%	Cox (multi)	OS	27/93	0.61	0.43	0.26~1.41	No	83%	No	Whole section	Not approved	Rabbit
Wang 2017 [24]	TC	5%	Cox (multi)	OS	26/81	1.3	0.27	0.76~2.21	No	76%	No	Whole section	Not approved	Rabbit
Zhang 2023 [25]	TC	1%	Cox (multi)	OS	81/212	1.54	0.21	1.03~2.32	Yes	N/A	No	Whole section	Approved	Mouse
Chen 2020 [13]	TC	1%	KM	PFS	21/100	0.5	0.27	0.29~0.84	No	91%	N/A	Whole section	Approved	Mouse
Darb-Esfahani 2015 [14]	TC	0 (any)	KM	PFS	153/177	0.47	0.33	0.24~0.9	No	N/A	No	TMA	Not approved	Rabbit
Kim 2018 [19]	TC	5%	KM	PFS	36/108	1.2	0.65	0.34~4.28	Yes	N/A	Yes	TMA	Approved	Rabbit
Bas 2021 [12]	CPS	5%	KM	OS	9/94	1.61	0.45	0.67~3.86	No	65%	N/A	TMA	Approved	Mouse
Chen 2020 [13]	CPS	1	KM	OS	48/100	0.93	0.25	0.57~1.52	No	91%	N/A	Whole section	Approved	Mouse
Eymerit-Morin 2021 [15]	CPS	1	KM	OS	22/59	0.64	0.34	0.32~1.25	No	60%	Yes	TMA	Not approved	Rabbit
Lee 2022 [20]	CPS	10	KM	OS	21/139	0.58	0.24	0.37~0.93	Yes	N/A	Yes	TMA	Approved	Mouse
Zhang 2023 [25]	CPS	1	KM	OS	127/212	1.17	0.16	0.85~1.60	Yes	N/A	No	Whole section	Approved	Mouse
Darb-Esfahani 2015 [14]	IC	20/mm^2^	Cox (multi)	OS	60/200	0.73	0.26	0.44~1.2	No	N/A	No	TMA	Not approved	Rabbit
de la Fuente 2020 [22]	IC	1%	Cox (multi)	OS	26/130	0.62	0.34	0.32~1.2	Yes	N/A	Yes	TMA	Approved	Mouse
Mills 2019 [23]	IC	1%	Cox (multi)	OS	70/93	1.43	0.38	0.68~3.0	No	83%	No	Whole section	Not approved	Rabbit

^1^ HR: hazard ratio, ^2^ SE: standard error, ^3^ CI: confidence interval, ^4^ NACT: neoadjuvant chemotherapy.

**Table 3 diagnostics-13-03258-t003:** Pooled results of the categorized data.

Survival Type	Scoring	Data Type	Number of Studies	HR ^1^	CI ^2^	*p*-Value	I ^2^	*p*-Value
OS ^3^	CPS ^5^	KM ^8^	5	0.9	0.64–1.26	0.53	54.1	0.07
OS ^3^	IC ^6^	Multi Cox ^9^	3	0.82	0.54–1.26	0.37	33.7	0.22
OS ^3^	TC ^7^	Multi Cox ^9^	4	0.93	0.54–1.62	0.81	73.5	0.01
OS ^3^	TC ^7^	KM ^8^	8	0.9	0.48–1.69	0.75	86.5	<0.0001
PFS ^4^	TC ^7^	KM ^8^	3	0.53	0.35–0.78	0.0015	0.0	0.42

^1^ HR: hazard ratio, ^2^ CI: confidence interval, ^3^ OS: overall survival, ^4^ PFS: progression-free survival, ^5^ CPS: combined positive score, ^6^ IC: immune cell, ^7^ TC: tumor cell, ^8^ KM: Kaplan-Meier estimate, ^9^ Multi Cox: multivariate Cox proportional hazard model.

**Table 4 diagnostics-13-03258-t004:** Subgroup analysis result.

Survival Type	Scoring	Data Type	Subgroup		Number of Studies	HR ^1^	CI ^2^	*p*-Value	I ^2^
OS ^3^	CPS ^4^	KM ^6^	FDA approval	Approved	4	0.96	0.65–1.40	0.82	59.1
				Not approved	1	0.64	0.33–1.25	0.19	
			Stage	High stage only	2	0.84	0.42–1.68	0.63	83.2
				High and low stage	3	0.92	0.64–1.32	0.65	27.0
			NACT	Included	2	0.6	0.40–0.88	0.009	0.0
				Not included	1	1.17	0.86–1.60	0.32	
			Ab species	Mouse	4	0.96	0.65–1.40	0.82	59.1
				Rabbit	1	0.64	0.33–1.25	0.19	
OS ^3^	TC ^5^	KM ^6^	FDA approval	Approved	5	0.73	0.37–1.44	0.36	89.0
				Not approved	3	1.43	0.34–6.02	0.62	82.9
			Stage	High stage only	2	1.58	0.89–2.80	0.12	65.2
				High and low stage	6	0.75	0.34–1.65	0.47	82.3
			NACT	Included	2	2.14	0.51–8.95	0.3	78.0
				Not included	5	0.8	0.38–1.71	0.57	89.5
			Ab species	Mouse	2	0.9	0.17–4.77	0.9	94.1
				Rabbit	6	0.9	0.43–1.89	0.78	82.0
			Region	Asia	4	1.1	0.49–2.45	0.81	85.8
				Europe	2	0.59	0.42–0.82	0.0017	5.4
				Africa	1	4.99	1.44–17.29	0.01	
				America	1	0.37	0.18–0.76	0.006	
			Slide type	Whole slide	4	0.88	0.35–2.25	0.8	87.7
				TMA	2	0.62	0.18–2.14	0.45	86.9

^1^ HR: hazard ratio, ^2^ CI: confidence interval. ^3^ OS: overall survival, ^4^ CPS: combined positive score, ^5^ TC: tumor cell, ^6^ KM: Kaplan-Meier estimate.

**Table 5 diagnostics-13-03258-t005:** Subgroup analysis result of FDA approved antibody data.

Survival Type	Scoring	Data Type	Subgroup		Number of Studies	HR ^1^	CI ^2^	*p*-Value	I ^2^	*p*-Value
OS	CPS	KM	Stage	High stage only	2	0.84	0.42–1.68	0.63	83.2	0.01
				High and low stage	2	1.08	0.67–1.75	0.74	13.5	0.28
OS	TC	KM	Stage	High stage only	2	1.58	0.89–2.80	0.12	65.2	0.09
				High and low stage	3	0.46	0.30–0.73	0.0009	49.3	0.13
			NACT	Included	1	1.14	0.66–1.98	0.64		
				Not included	3	0.86	0.68–1.08	0.61	93.3	<0.0001
			Ab species	Mouse	2	1.36	0.96–1.92	0.9	94.1	<0.0001
				Rabbit	3	0.65	0.32–1.21	0.16	74.6	0.02

^1^ HR: hazard ratio, ^2^ CI: confidence interval.

## Data Availability

The datasets generated during and/or analyzed during the current study are available from the corresponding author upon reasonable request.

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
