# Peer review of "Prognostic Significance of Programmed Cell Death Ligand 1 Expression in High-Grade Serous Ovarian Carcinoma: A Systematic Review and Meta-Analysis"

_diagnostics, 2023, doi:10.3390/diagnostics13203258_

Round 1

Reviewer 1 Report

Dear Authors,

The article is interesting. Ovarian cancer still represents a challenging diagnosis and overall management nowadays, still associating a poor prognosis. Thus new markers of prognostics are very useful. The meta-analysis is well designed and well presented, the style is impeccable and the research adds value to our current knowledge. I only have a few minor comments.

Here are my observations:

1.       Title – no need for point at the end of the title.

2.       Abstract - no need for numbers with concern to each section since these sections are standard.

3.       Abstract. Please provide a few core data (numbers, statistics) based on your research starting for the published data.

4.       Introduction. Please specify the current role of PD-L1 assessments in other cancers (the most practical aspects nowadays).

5.       Table 1. Reference number should be included within the first column.

6.       Table 1. Is the display is based on the data of publication or another criteria has been used?

7.       Results. Please use “p” (not capital letter) for p-value.

8.       Conclusion. Please avoid “In summary” after announcing “5.Conclusions”

Thank you

Author Response

===================================================

Reviewer 2 Report

Although PD-L1 expression has been discussing intensively related to different type of tumours recently, the potential benefit of this biomarker e aluatiom in high grade serous carcinoma has yet to be established. This systematic review and meta- analysis is comprehensive and well organised, including many available investigations, either whole section evaluation or TMA. At the same time a conclusion is not so impactful because the relationship between positive PD-L1 expression and good prognosis is not surprising. It would be more interesting to discuss PD-L1 expression and HGSC immunoreactive molecular subtype, because molecular subtyping is a futures step for patients subdividing, and more personalised therapy. 
